# Generic Chemometric Models for Metabolite Concentration Prediction Based on Raman Spectra

**DOI:** 10.3390/s22155581

**Published:** 2022-07-26

**Authors:** Abdolrahim Yousefi-Darani, Olivier Paquet-Durand, Almut Von Wrochem, Jens Classen, Jens Tränkle, Mario Mertens, Jeroen Snelders, Veronique Chotteau, Meeri Mäkinen, Alina Handl, Marvin Kadisch, Dietmar Lang, Patrick Dumas, Bernd Hitzmann

**Affiliations:** 1Department of Process Analytics und Cereal Science, Institute for Food Science and Biotechnology, University of Hohenheim, Garbenstr. 23, 70599 Stuttgart, Germany; rahim@uni-hohenheim.de (A.Y.-D.); almut.vonwrochem@uni-hohenheim.de (A.V.W.); bernd.hitzmann@uni-hohenheim.de (B.H.); 2Bayer AG, L Kaiser-Wilhelm-Allee 1, 51373 Leverkusen, Germany; jens.classen@bayer.com (J.C.); jens.traenkle@bayer.com (J.T.); 3Sanofi, Cipalstraat 8, 2440 Geel, Belgium; mario.mertens@sanofi.com (M.M.); jeroen.snelders@sanofi.com (J.S.); 4Department of Industrial Biotechnology, School of Engineering Sciences in Chemistry, Biotechnology and Health, Royal Institute of Technology (KTH), 109 06 Stockholm, Sweden; chotteau@kth.se (V.C.); meerim@kth.se (M.M.); 5Rentschler Biopharma SE, Erwin-Rentschler-Street 21, 88471 Laupheim, Germany; alina.handl@rentschler-biopharma.com (A.H.); marvin.kadisch@rentschler-biopharma.com (M.K.); dietmar.lang@rentschler-biopharma.com (D.L.); 6GSK, Rue de l’Institut 89, 1330 Rixensart, Belgium; patrick.dumas@gsk.com

**Keywords:** generic model, Raman spectroscopy, on-line process monitoring, PLS regression, chemometrics, CHO cell cultivation

## Abstract

Chemometric models for on-line process monitoring have become well established in pharmaceutical bioprocesses. The main drawback is the required calibration effort and the inflexibility regarding system or process changes. So, a recalibration is necessary whenever the process or the setup changes even slightly. With a large and diverse Raman dataset, however, it was possible to generate generic partial least squares regression models to reliably predict the concentrations of important metabolic compounds, such as glucose-, lactate-, and glutamine-indifferent CHO cell cultivations. The data for calibration were collected from various cell cultures from different sites in different companies using different Raman spectrophotometers. In testing, the developed “generic” models were capable of predicting the concentrations of said compounds from a dilution series in FMX-8 mod medium, as well as from an independent CHO cell culture. These spectra were taken with a completely different setup and with different Raman spectrometers, demonstrating the model flexibility. The prediction errors for the tests were mostly in an acceptable range (<10% relative error). This demonstrates that, under the right circumstances and by choosing the calibration data carefully, it is possible to create generic and reliable chemometric models that are transferrable from one process to another without recalibration.

## 1. Introduction

For over 30 years now, Chinese hamster ovary (CHO) cells have been used in the production of recombinant protein pharmaceuticals [1]. In addition, during this time span, the productivity of CHO cultivation processes increased drastically. However, with these improvements also came ever increasing requirements in monitoring and controlling these processes properly. Rapid online monitoring systems based on spectroscopy and multivariate statistics are one possibility to further increase process monitoring capabilities.

Raman spectroscopy provides detailed chemical information and is routinely used in various application areas, including pharmaceutical industries [2,3,4,5,6,7,8,9,10]. 

Spectroscopic data consist of thousands of variables (wavenumbers) and measurements (objects/observations). To utilize the complete information of the complex spectra and to handle the large dataset, multivariate analysis is needed. The main aim of these statistical analysis techniques is to perceive the relationship between the variables. This is based on the idea of considering many nonselective variables instead of just one variable, and then ultimately combining them in a multivariate data-driven prediction model [11]. 

The combination of statistical data analysis with Raman spectra could lead to on-line determination of important process parameters and critical quality attributes, enabling nondestructive on-line monitoring and process control. Recently, several authors have used spectra collected from in-line Raman probes, coupled with multivariate data-driven prediction models for on-line monitoring of cell culture variables, such as glucose and lactate [12,13]. For a detailed exposition on implementation of prediction models coupled with Raman spectroscopy in cell culture processes, the reader is referred to the literature [14,15,16].

Despite the successful application of such prediction models, they have severe problems regarding flexibility in system specification, meaning that these methods and models need to be recalibrated whenever the system that they are applied to changes even slightly. A model or method to measure a particular variable cannot be transferred from one spectrometer to another in most cases. In this context, development of generic models would be a helpful solution. 

A generic chemometric prediction model that can predict variables of interest from spectroscopic data is difficult to create. Chemometric models are usually data-driven statistical models that rely on correlations between the spectroscopic data and the variable of interest. Any tiny variation in process setup, spectroscopic hardware or settings, or general measurement and process condition could break these correlations. Then, the model would no longer be accurate.

However, if the spectroscopic method has a high specificity, such as Raman or NIR spectroscopy, there might be signals in these spectra that correlate to the target variable not just by some unknown indirect effects, but they are directly caused by the variable of interest (or the causality chain from the variable of interest to the spectral information is at least very short). If that is the case, these spectroscopic methods might be useful for a generic modeling approach. Still, there is a lot of other information in the spectra that might interfere with the model. However, this distracting information, such as noise, baseline shift, and scattering, can mostly be removed by signal preprocessing techniques.

Development of generic chemometric models for prediction of cell line process variables have been recently reported in the literature [17,18,19]. Most of these studies are based on model building and prediction based on data obtained from the same Raman spectrometer, leaving aside instrumental variability, which is a key issue when using Raman spectroscopy in biopharmaceutical environments [20].

Thus, the present contribution is dedicated to the improvement of Raman-based generic calibration models using a large and diverse Raman dataset combined from several mammalian cell culture batches obtained from four industrial sites. These cultivations were run under various process conditions using different cell lines, covering variations in process metabolites, cell growth, and product concentration.

Generic chemometric models for predicting glucose, lactate, glutamine, and glutamate concentrations were made from the datasets and were validated using two different independent datasets, which were not used in the calibration of the models. The selected variables were the “least common denominator”, meaning they were measured and available in all datasets from all different sources. 

Such a generic calibration method is highly desired, since, for evident cost reasons, it becomes difficult to develop a new spectroscopic predictive model any time you want to explore its potential considering a new Raman spectrometer or a new cell line. Furthermore, a calibration model that is independent from the spectral influence of the applied Raman spectrometer would significantly enlarge the application possibilities of Raman spectroscopy.

## 2. Materials and Methods

### 2.1. Calibration Dataset

Spectra were taken on-line during cultivations of various producer CHO cells (not exclusively from the Rentschler suspension CHO producer cell line) for the production of biopharmaceuticals from 4 different industrial sites, i.e., 4 different companies. A total of 1699 data points (Raman spectra and according off-line information) were available. In total, there were 305 spectra from site 1 acquired from 6 different batches, 958 spectra from site 2 acquired from 22 different batches, 295 spectra from site 3 acquired from 4 different batches, and 148 spectra from site 4 acquired from 7 different batches. The following spectrometers were used to acquire the information:

Site 1 and 2: Kaiser optical systems, RXN2 (laser wavelength 785 nm), wavenumbers: 100–3425 cm^−1^, increment per 1 cm^−1^. 

Site 3 and 4: Resolution Spectra Raman Spectrometer (laser wavelength 785 nm), class 4 laser, 100–4000 cm^−1^, increment per 3 cm^−1^. 

Off-line measurements were performed by Cedex Bio analyzer (Roche Diagnostics, Basel, Switzerland).

### 2.2. Validation Dataset

For validation and testing purposes two different datasets were used:

1. Raman spectra obtained from a dilution series of glucose, lactate, glutamine, and glutamate in FMX-8 mod medium, where FMX-8 (Cell Culture Technologies LLC, Gravesano, Switzerland) mod medium was supplemented with L-alanine, L-aspartic acid, L-glutamic acid, pluronic-F68 and D (+)-glucose, as described in the literature [21]. The following concentrations were prepared in triplicate: 100, 50, 40, 30, 20, 11, 9, 7, 5, and 3 g/L. Raman spectra of these solutions were measured using a Tec5/InnoSpec Raman 785 nm spectrometer with an InPhotonics RPS785/FF12 process probe with the focus lens removed. Removing the 6 mm focal lens decreased the signal intensity by a factor of ~10, but it increased the reproducibility by removing the strict measurement distance constraint. The clear samples were measured with 2.5 s integration time in 20 mL flasks wrapped in aluminum foil to exclude any environmental light.

2. Independent dataset from a new CHO host cell cultivation obtained from the Royal Institute of Technology in Stockholm (KTH). The Chinese hamster ovary suspension cell line recombinantly producing an IgG1 antibody kindly provided by Rentschler Biopharma SE (Laupheim, Germany) was mostly used. A fed-batch cultivation was performed with proprietary base medium and feed medium, BM/FM. Glucose and glutamine were daily fed according to the cell needs, based on measured consumption rate of these substrates. Sampling was performed daily, and the concentrations of glucose and lactate were measured by Cedex Bio analyzer (Roche Diagnostics). A Raman RXN2 Analyzer with immersion optics probes (Endress+Hauser Optical Analysis, Inc, Ann Arbor, MI, USA) was mounted on the bioreactor (Belach Sweden) and used with laser excitation wavelength 785 nm and spectral coverage of 150–3425 cm^−1^ (Raman shift). Raman spectra were recorded during the culture; 75 spectra were captured and averaged per single recorded spectrum with 10 s exposure time each, resulting in a recording interval of ≈12.5 min.

### 2.3. Preprocessing

Due to variations in optical pathways of the spectrometers used by each site, spectra will be projected on different pixels. Therefore, prior to performing multivariate data analysis, an evaluation method for standardization of the wavenumber is crucial. 

The spectral range that is available on all spectrometers and represents the least common denominator starts at 300 cm^−1^ and goes up to 3400 cm^−1^ with an increment of 3 cm^−1^. 

Since a unified data structure is required, where all spectra have the same length and the same wavelengths are at the same relative position, a cubic spline interpolation in this wavenumber range was performed on all acquired Raman spectra. With the application of such a standardization procedure, the collected Raman spectra are recalculated as if they were all collected with the same spectrometer (assuming that the spectrometers are reasonably well-calibrated). 

In the next step, combinations of different preprocessing methods were assessed to evaluate the best possible combination, leading to maximum signal-to-noise ratio. The preprocessing combination, the first derivative calculated by Savitzky–Golay (SG) (2nd order polynomial, 31-point fitting window), followed by standard normal variate (SNV) normalization, was chosen for model calibration.

The goal of the Savitzky–Golay first derivative preprocessing is to remove backgrounds and reduce the noise content of the spectra by smoothing and numerically increasing the apparent resolution of the spectra by a sharpening of the zero-order spectra peaks [15]. The SNV preprocessing removes a constant offset term in the spectra, since it normalizes each value of the spectra by the standard deviation (SD) of pooled variables, thus bringing all spectra to the same scale with a unit SD [20].

Figure 1 shows Raman spectral data from a number of collected samples, before (A) and after (B) preprocessing. Figure 1B indicates the baseline drift can be removed using the applied preprocessing methods. 

### 2.4. Model Development

First, in-line Raman spectra were matched with their respective off-line measurements and, after applying the preprocessing methods, partial least squared regression (PLS-R) models were constructed. PLS-R is a multivariate statistical method that aims to establish a model that relates the variations of the spectral data to a series of relevant targets. The spectral data (X matrix) are, thus, related to the targets (Y matrix) according to the linear equation Y = XB + E, where B is a matrix of regression coefficients and E is a matrix of residuals [22]. Here, the (X) matrix included the preprocessed spectra and the (Y) matrix is composed of the molecule concentrations measured off-line (glucose, lactate, glutamine, and glutamate).

Separate PLS-R models were developed for prediction of glucose, lactate, glutamine, and glutamate. A cross-validation procedure was performed in order to assess the optimal number of latent variables for each regression model. The root mean square error of cross-validation (RMSECV) was calculated following the RMSE formula given in Equation (1):(1)$$\mathrm{RMSE}=\sqrt{\frac{1}{N}\overset{N}{\underset{i=1}{\sum }}{\left({\hat{Y}}_{i}-Y_{i}\right)}^{2}}$$
where $\hat{Y_{i}}$ represents the predicted value, $Y_{i}$ is the measured (off-line) value, and N stands for the measurement count.

The performance of each model was assessed using correlation coefficient (R^2^), root mean square error of prediction (RMSE), and the standard error of prediction (SEP).

R^2^ represents the ration ratio of variability captured by the model to the total variability, RMSE provides an estimate of the prediction error in the same unit as the initial data (g L^−1^), and SEP and SEC provide the prediction and calibration error with respect to the concentration range of the reference data in terms of percentage. SEC and SEP are calculated as follows:(2)$$\mathrm{SEC}\;\mathrm{or}\;\mathrm{SEP}\left(\%\right)=\;\frac{\mathrm{RMSE}}{Y_{\mathrm{range}}}\;\times 100\%$$

$Y_{\mathrm{range}}$ is the range of the concentration in the reference data (the highest minus the smallest value). In bioprocess monitoring, SEP values of less than 5% are usually considered very good and values above 10% are usually considered too high/not acceptable. 

## 3. Results and Discussion

### 3.1. Model Calibration

Off-line data from 61 cell cultivation runs containing 1699 Raman spectra from four different industrial sites were combined with their respective spectra and used to produce generic prediction models for glucose, lactate, glutamine, and glutamate. The dataset was split into a training set (70%) and test set (30%). The training set was used for model calibration and the prediction accuracy of the models have been evaluated using the test set. As all industrial sites involved in this study prefer to keep the concentration of the components used in their cultivations undisclosed, the reference values and the predicted values are scaled to a range from 0 to 1. 

Figure 2 illustrates scaled model predictions plotted against scaled reference measurements for the calibration dataset for glucose (a), lactate (b), glutamine (c), and glutamate (d), as well as the model identification and results, specifically number of latent variables (LV), values of coefficient of determination (R2), and standard error of calibration or root mean squared error (SEC%), which is the same in this case due to the normalization. 

It was observed that developed models were performing with acceptable error, having high R2 values (above 0.93 for all models) and low SEC% values (2.4%, 3.0%, 2.5%, and 5.9% for glucose, lactate, glutamine, and glutamate, respectively). Additionally, all models had a relatively low number of latent variables, apart from glutamate. The high number of latent variables for the glutamate model may indicate that the model is over-fitted for this parameter and may not be robust enough when applied to independent data.

Calibration plots for predicted glucose, lactate, and glutamine indicate that, in general, all models correlate well with the measured values. The exceptions are (i) low concentrations of glucose samples for the glucose model and (ii) high concentrations of lactate samples for the lactate model. Having a low number of samples in these ranges affects the model predictions. Although model fit is less favorable for these samples, it was investigated whether the presence of these samples in the dataset improved the accuracy of model predictions. 

Figure 3 illustrates model validation results using the scaled test set plotted against scaled reference measurements for glucose (A), lactate (B), glutamine (C), and glutamate (D) models.

### 3.2. Model Validation with Spectra Acquired from Dilution Series of the Main Compounds in FMX-8 Mod Medium

The main goal of this study was to develop a generic model for predicting important process variables of CHO-based cell cultivations, which would be sufficient and robust for measuring these process variables in various different media with different spectrometers. As mentioned in the previous section, generic calibration models were developed by using calibration datasets from four different industrial sites. In order to challenge these generic models with new datasets (external validation), spectra from a dilution series of glucose, lactate, glutamine, and glutamate in FMX-8 mod medium were acquired with a different Raman spectrometer.

The external validation results of the generic model were compared with the external validation results of individual calibration models made from datasets of each site. To avoid promoting one site’s instrument or calibration dataset over another, we name the industrial sites’ datasets randomly from 1–4. The preprocessing methods for the individual calibration models are similar to the ones used for the generic models. Datasets were split to training set (70%) and test set (30%), and PLS regression models were made. The errors of calibration (training set) and test set, as well as the error of external validation, for the glucose prediction models (from the individual models) are presented in Table 1.

As can be seen in Table 1, except for site 3, the glucose models had relatively low values of SEP and SEP for the training set, as well as the test set, indicating reasonable accuracy of the models. However, for all individual models, the errors of external validation were quite high. This was due to the fact that the transferability of the models to a new dataset from a new instrument was not considered, such as the Raman shift which would occur when measurements are performed with different instruments. Similar results were obtained for lactate, glutamine, and glutamate models, and are not presented here.

The use of a generic model is an efficient approach to calibration transfer when instrument-to-instrument performance variability is anticipated. Therefore, the developed generic models, which were obtained by combining the data from the four sites into a single dataset, were used for predicting the concentration of variables in the FMX-8 mod medium. Results are presented in Figure 4.

The generic models developed for glucose, lactate, and glutamine were found to be satisfactory in predicting concentrations of these compounds in FMX-8 mod medium with SEP of 4.6, 6, and 5%, respectively. However, higher errors were obtained at higher concentrations. These relatively high errors were mainly due to the concentration range used in the calibration set. Indeed, it is well known that PLS regression should not be used for extrapolation, and we should not overlook the fact that predictions are not accurate when concentration ranges of a calibration set and validation sets are not representative of each other. However, in an actual application of such a model in a bioprocess, encountering such high concentrations (i.e., >50 g/L of glucose) is very unlikely and, therefore, higher errors at higher concentrations should not be overemphasized.

The generic model developed for glutamate was unable to accurately predict the concentrations of glutamate in FMX-8 mod medium, as the SEP was much higher than 10% (SEP = 16%). As discussed previously, this was somewhat expected and could be related to the high number of principal components required in the calibration model, this generic model would not be robust enough when applied to independent data.

### 3.3. Model Validation with CHO-Based Cell Cultures

In order to evaluate the applicability of the generic models to new cell lines based on CHO host cells, the generic models (glucose and lactate) have been used to predict metabolite concentrations of a new CHO cell cultivation, which was not presented in the calibration dataset.

Figure 5 illustrates model predictions versus reference measurements and corresponding RMSE values of the lactate and glucose models for this validation batch.

It is shown that the predicted profiles followed the trends of the metabolites observed using reference measurements. However, an SEP value of 64% has been obtained for glucose, due to a large overprediction of ~1.5 g/L. This was likely due to the relatively narrow concentration range for glucose (2–4 g/L) in comparison to the calibration range (0–20 g/L). Inclusion of more off-line reference measurements covering this range of glucose concentrations should allow for a more robust model to be developed with an improved ability to monitor changes in glucose concentration [23]. Furthermore, glucose predictions are specifically more accurate when high glucose concentrations are present, which is consistent with what was observed previously with the original calibration dataset (Figure 2A). 

In contrast to the glucose model, the lactate model predicted with excellent accuracy throughout the run (SEP of 4%), closely following trends of lactate changes observed using the reference measurement method (Figure 5B).

Overall, the independent validation results of the generic models indicate that this approach seeks to incorporate the variance associated with the transfer into the data used for calibration model development. A generic modeling approach relies on collecting calibration data that reflect all future sources of variance [20,23]. This can be achieved by including calibration data from different cell lines obtained from different instruments over different operating conditions to introduce all expected variability arising from instruments and differences in the cultivation processes.

## 4. Conclusions

A set of generic calibration models based on Raman spectra were developed and used to predict glucose, lactate, glutamine, and glutamate concentrations. The calibration dataset included spectra from multiple batches of CHO-based cell lines obtained from four different industrial sites, which included changes in proprietary media as well. PLS regression models were used to correlate spectra with reference measurements. Models were validated using two different independent validation datasets, which were not present in the calibration datasets:

(1)Dilution series of glucose, lactate, glutamine, and glutamate in FMX-8 mod medium measured with a different spectrometer (Tec5 RAMAN 785 spectrometer) obtained from university of Hohenheim.(2)Independent dataset from a new CHO host cell line cultivation obtained from KTH.

The results of this study indicate that the developed generic models for glucose, lactate, and glutamine are able to predict these compounds in the FMX-8 mod medium within an acceptable accuracy (SEP < 6%). The prediction error was found to be relatively high for glutamate predictions (SEP = 16%). This could be due to the concentration range of glutamate in the calibration set, which was of a few milli- or micromoles per liter, making it difficult to generate an accurate prediction model. Another possible reason could be that, with the applied preprocessing methods, the identified Raman peaks are too weak and insignificant. These should be considered for future improvements of the model.

Most important to note, however, is the ability of the glucose and lactate generic models to be applicable to a new cell line which is not considered in the calibration dataset, providing evidence that the generated models are independent of scale and applicable to new cell lines derived from CHO host cells. It is important to note that the accuracy of the generic glucose model is limited at lower concentration ranges and, hence, in these ranges, models can only be used to capture trends in concentration changes rather than for measurement of absolute values.

Overall, the development of such a generic model is a valuable asset, as it has the potential to provide reliable predictions of variables of interest. Even if no transfer of the model from one system to another is planned, a generic model is also much more robust to unwanted changes of process conditions, which would otherwise be a problem for chemometric models.

## Figures and Tables

**Figure 1 sensors-22-05581-f001:**
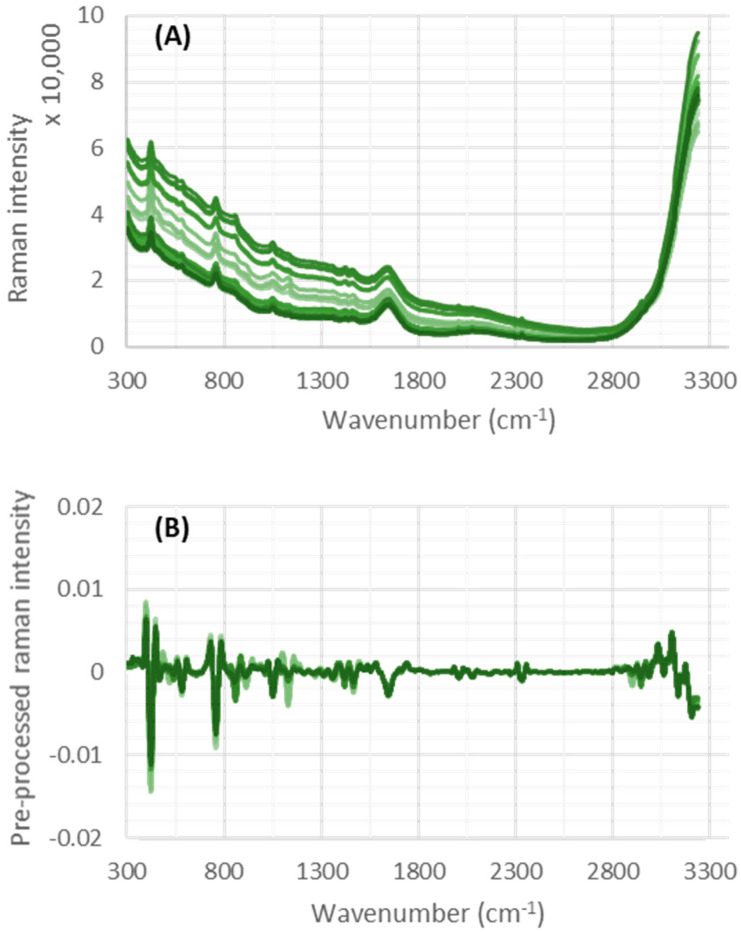
(**A**) Raman spectra acquired from a single cell culture. (**B**) Spectra after preprocessing revealing a baseline correction and providing relevant Raman contribution.

**Figure 2 sensors-22-05581-f002:**
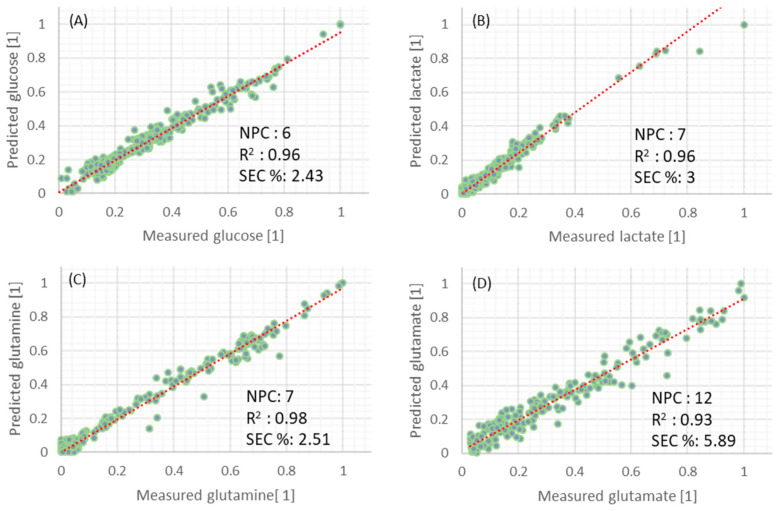
Normalized prediction results of glucose (**A**), lactate (**B**), glutamine (**C**), and glutamate (**D**) from the training set. Dashed red lines are the ideal model fit (1:1).

**Figure 3 sensors-22-05581-f003:**
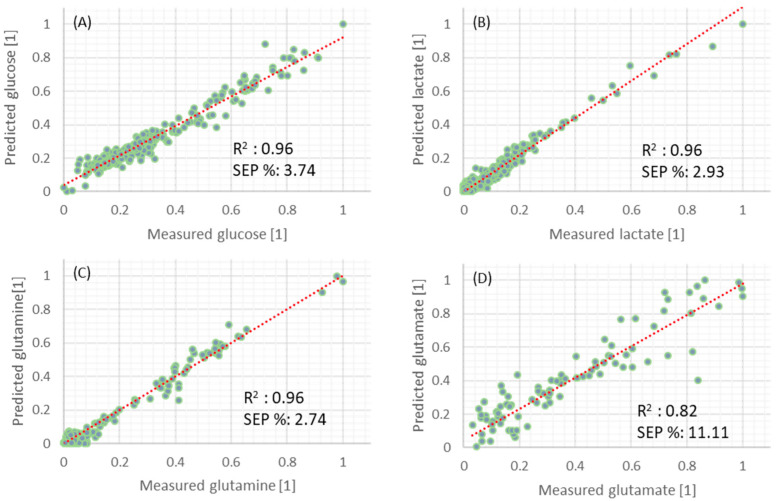
Normalized prediction results of glucose (**A**), lactate (**B**), glutamine (**C**), and glutamate (**D**) from the test set. Dashed lines are the ideal model fit (1:1).

**Figure 4 sensors-22-05581-f004:**
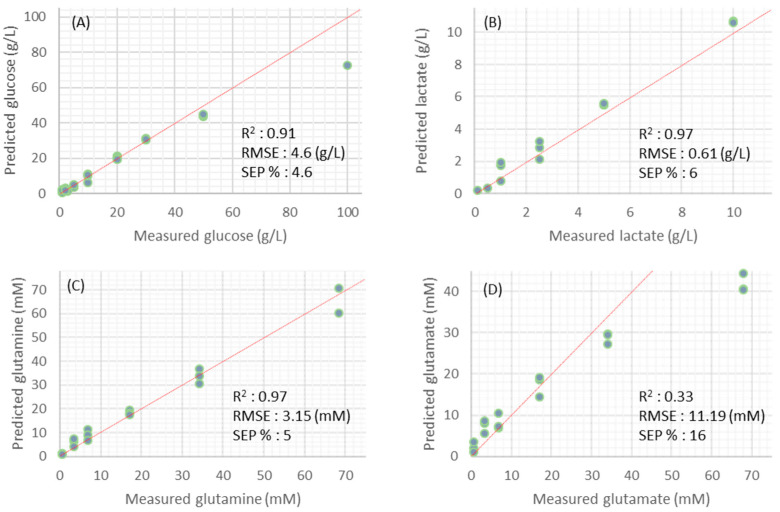
Predicted vs. off-line values obtained by the generic models (glucose (**A**), lactate (**B**), glutamine (**C**), and glutamate (**D**)) on spectra obtained from a dilution series of the compounds in FMX-8 mod medium. Dashed lines are the ideal model fit (1:1).

**Figure 5 sensors-22-05581-f005:**
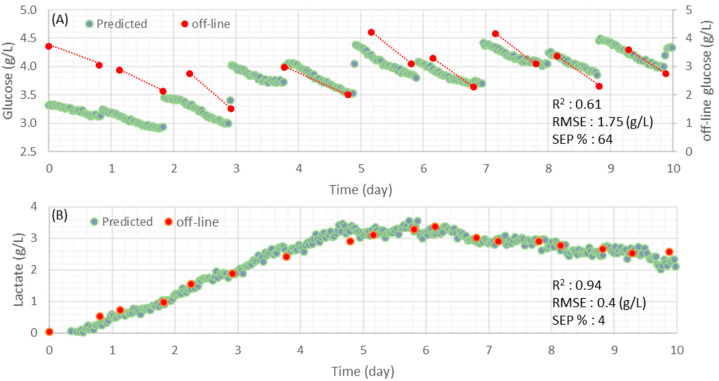
Prediction obtained by the glucose generic model (**A**) and the lactate generic model (**B**) on spectra acquired from an independent CHO cell fed-batch cultivation performed with BM/FM. Dashed lines are presented to make it easier to follow the general trend. Similar predictions for glutamine and glutamate are not shown, as their concentrations were too low (<5 mmol/L) to be predicted accurately.

**Table 1 sensors-22-05581-t001:** Glucose prediction models from individual sites.

	Calibration (70%)		Test Set (30%)		External Validation	
Data Set	R^2^	SEC%	R^2^	SEP%	R^2^	SEP%
**Site 1**	0.73	3.01	0.69	4.11	0.67	11.7
**Site 2**	0.97	2.73	0.93	4.57	0.61	17.6
**Site 3**	0.92	5.36	0.47	12.2	0.52	22.1
**Site 4**	0.94	5.45	0.92	5.95	0.67	12.7

## Data Availability

Not applicable.

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
