# Peer review of "Generic Chemometric Models for Metabolite Concentration Prediction Based on Raman Spectra"

_sensors, 2022, doi:10.3390/s22155581_

Round 1

Reviewer 1 Report

The authors proposed a possible method to establish the generic chemometric model for metabolite  concentration prediction based on Raman spectra. It is useful  to on line monitor in pharmaceutical  bioprocesses using Raman spectra. However, my questions are 

1. In line 268, the article described "The generic model developed for glutamate was unable to accurately predict the oncentrations of glutamate in FMX-8 mod medium (SEP = 16 %)", why the manuscript expressed the SEP<10% ( in abstract line 27) ?  Suggest to give a rational explanation.

2. In section "Model validation with CHO based cell cultures ", why authors did not  provided  the predicted profiles of  glutamine  and glutamate  like glucose and lactate, Is it due to not accuracy or other reasons? 

3. There is  word spelling error, such as  "various cell cultures form different sites" in line 22 , the "form" should be "from "

Author Response

Thank you for your helpful sugestions and remarks.

We changed the manuscript slightly to adress the issues mentioned:

  1. We added "moslty" to the abstract. Before we were stating that all results were better than 10 % SEP which was obviously not correct. We also added a statement to Material and Methods stating that SEP smaller 5 % are very good and SEP bigger 10 % are usually not acceptable. 
  2. For the experiment in question, the prediction of glutamate and glutamine were not not shown as they were not accurate. The concentrations of both compounds were in the 1-5 mmol/L range which was just too low. We added an according statement to the caption of Figure 5.
  3. We corrected the spelling error

Kind regards,

Olivier Paquet-Durand

Reviewer 2 Report

The paper by Yousefi-Darani and coworkers presents an attractive multi-centre chemometric model based on Raman spectroscopy for measuring metabolites commonly used in cell cultures.

The paper deserves publication; however, I have some suggestions for the authors:

1) In the INTRODUCTION, please provide some information about the interest in using the CHO- cell cultures and the 4 chosen metabolites.

2) Please give more details about the spline interpolation line before pre-processing. I also suggest being more cautious about the fact that this procedure would eliminate any difference between spectrometers.

3) Please clarify if the first derivative procedure already integrates the Savitzky-Golay smoothing and justify the 31 points fitting (are there any references about such an extensive smoothing range?)

4) Please make some additional comments on the low R2 and high SEP % obtained for the external validation

5) Please add some comments on the concentration ranges far from the prediction model (i.e. high concentration glucose): are they expected in the lab practice?

Minor comments:

1) In the ABSTRACT, rewrite the sentence in line 19 (We could demonstrate that the last statement is not strictly true); and define the acronym SEP.

2) Is there any particular reason for the different line colours in figure 1?

3) In line 164, you have reference [232].

4) If SEP and SEC are used as synonyms, please choose only one denomination to avoid confusion among the readers.

5) What is the meaning of [1] in the X-axis in figures 2 and 3?

6) In line 270: "principle" was used instead of "principal"?

7) In figure 5, "predicted" and "off-line" values seem to have the wrong colour.

Author Response

Thank you for your helpful suggestions and comments!

We changed parts of the manuscript to address your questions and comments:

Suggestions:

  1. We focused on the four variables (glucose, lactate, glutamine, glutamate) simply due to data availability. From all the data sets from different sources, these four variables were the ones that were always measured/available. These are also (arguably) the most important variables, when monitoring mammalian cell cultures. We also added a small indtroductory section about CHO cell cultures.
  2. We changed the section slightly to hopefully clarify! We used a cubic spline interpolation here because it gave slightly better results, but to be honest the difference compared to a linear interpolation was not really significant. Of course, this alignment does not eliminate all the differences between the spectra, but it is simply necessary to get a unified data structure were all spectra have the same length and the same wavelength are at the same relative position.
  3. We changed the section slightly to clarify! The Savitzky Golay method can be used to smooth a signal but it can also be used to calculate a smoothed derivative of a signal by picking the appropriate coefficients. That is what we did. The window width of 31 points might seem excessive but since we pooled a lot of data from different sources together and some of the spectra were quite noisy, this was a necessary choice even if we might have lost some useful information due to oversmoothing.
  4. We added a sentence clarifying, why the error for glucose is so large: Although the general trend of glucose concentration over time was captured reasonably well, the model overpredicted the glucose amount by roughly 1.5 g/L. This large offset lead to the rather large SEP of 64 %.
  5. We added a short section to discuss this topic. It is in fact unlikely to encounter such high concentrations in actual bioprocesses. These results at very high concentrations should only be seen as a general test for the predictive capability of the generic model (or lack thereof).

Minor Comments:

  1. Since the sentence is more or less a "filler" and not really required, we just removed it.
  2. Since there are a lot of spectra in the plot, we used more colors. In principle the colors are not that important here and a black and white (or green and white) figure would suffice. We changed the figure accordingly.
  3. Corrected to "[23]"
  4. SEP and SEC are not used as synonyms. They are calculated the same way, but SEC is used for calibration errors and SEP is used for validation errors.
  5. [1] is the unit of measurement. We are not allowed to disclose the actual concentrations. The shown concentrations were therefore normalized to the respective max value.
  6. Typo: Corrected to principal
  7. We checked the figure. The colors seem to correct? 

Kind regards,

Olivier Paquet-Durand

This manuscript is a resubmission of an earlier submission. The following is a list of the peer review reports and author responses from that submission.